# Multidimensional poverty reduction effect of climate smart agriculture practices among rural households in Siltie Zone, Central Ethiopia

Jemil Yasin Shifa[1,2]* Abrham Seyoum Tsehay[1]

**1** Center for Rural Development, Addis Ababa University, Addis Ababa, Ethiopia, **2** Department of Rural Development and Agricultural Extension, Werabe University, Werabe, Ethiopia

* yasin.jemil73@gmail.com, jemilyasin@wru.edu.et

## Abstract

This study investigates the effect of Climate-Smart Agriculture adoption on multidimensional poverty reduction in the Siltie Zone, Central Ethiopia. Despite growing attention to CSA as a pathway toward sustainable development and poverty reduction, limited empirical evidence exists about its effect on multidimensional poverty. Using cross-sectional data collected from 416 smallholder farmers selected employing multi stage sampling procedure and GSEM, 2SLS and PSM analytical approaches by address potential endogeneity between CSA adoption and MPI. The multidimensional poverty index was constructed to capture deprivations across different dimensions while CSA adoption was measured based on the adoption and intensity of practices implemented. Results reveal that CSA adoption significantly reduces multidimensional poverty. These findings underscore the transformative potential of CSA in enhancing rural multidimensional poverty and highlight the importance of targeted policy interventions that promote different CSA practices adoption as a strategy. The study contributes to the growing body of literature linking sustainable CSA practices to inclusive and multidimensional poverty reduction as development outcomes.

## Introduction

Agriculture remains the backbone of Ethiopia's economy and contributes significantly to both the national GDP and rural livelihoods with over 70% of the workforce employed, however, the sector is increasingly threatened by climate change, erratic rainfall, and land degradation, which disproportionately affect smallholder farmers and keep them in a state of multifaceted poverty. Climate-Smart Agriculture (CSA) refers to an approach that sustainably increases agricultural productivity, enhances resilience to climate change, and reduces greenhouse gas emissions where possible [1]. CSA has emerged as a viable approach to address these problems [2,3].

**Data availability statement:** The data used for this study is publicly available at https://doi.org/10.6084/m9.figshare.31889473.

**Funding:** The author(s) received no specific funding for this work.

**Competing interests:** The authors have declared that no competing interests exist. Addis Ababa University Institutional Review Board (IRB), reference number: number:075/04/2024. This does not alter our adherence to PLOS ONE policies on sharing data and materials.

Varies studies supports the contribution of CSA technologies for the improvement of food security and agricultural productivity however, little is known about how it impacts multifaceted poverty in general and especially in developing nations like Ethiopia where poverty is pervasive across different locations and agroecological zones [4–6]

A numbers of previous studies focused on Food security effects of CSA adoption, but this study analyze MPI effects of different CSA practices adoption because it captures multiple dimensions of deprivation, including education, health, and living standards. Unlike income-based measures, MPI provides a more comprehensive assessment of household welfare, particularly in rural settings where non-monetary deprivations are significant. Furthermore, as shown Fig 1, this study addressed the potential endogeneity between CSA adoption and poverty outcomes through employing different analytical models including GSEM (generalized structural equation model), PSM (propensity square matching) and 2SLS (two stage least square) method of analysis. This study aims to fill these gaps by examining the causal relationship between CSA adoption and multidimensional poverty reduction in the study area using a cross-sectional survey. Therefore, this study provides evidence for policy makers and practitioners to create target climate smart agriculture technologies scale up and equitable interventions across different locations.

## Literature review

### Theoretical literature

It is necessary to establish theoretical frameworks in order to comprehend the connection between the adoption of CSA technologies and poverty reduction.

**Sustainable livelihoods framework (SLF).** Sustainable Livelihoods Framework (SLF), which offers a comprehensive lens that, shows the way rural households use various resources/ livelihoods, vulnerability contexts, policies institutions and procedures, livelihood strategies and livelihood outcomes [7]. CSA technologies as one of strategies households used to respond to climate change shocks and maintain their economic and social wellbeing since it increases resilience to climate-related risks influencing productivity and adaptability [3,6]

**Technology adoption theories.** The Diffusion of Innovations Theory [8] suggests that adoption is a social process influenced by the perceived attributes of the technology, communication channels, and social system. Furthermore, economic constraints model states that adoption decisions are influenced by factors such as income, credit access, and other factors.

**Capability approach and multidimensional poverty.** The Capability Approach, developed [9], shifts the focus of poverty analysis from income to capabilities. The model states that individuals have to live the life they value and measures poverty in different dimensions including health, education, and living standards. Thus CSA practices can be a vehicle to increase productivity and adaptation to climate shocks consequently leading to reducing poverty.

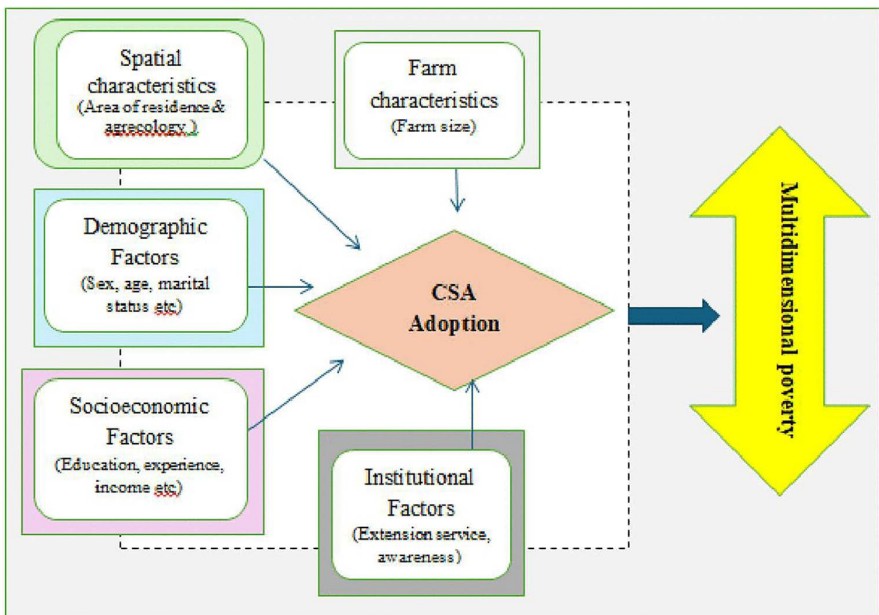

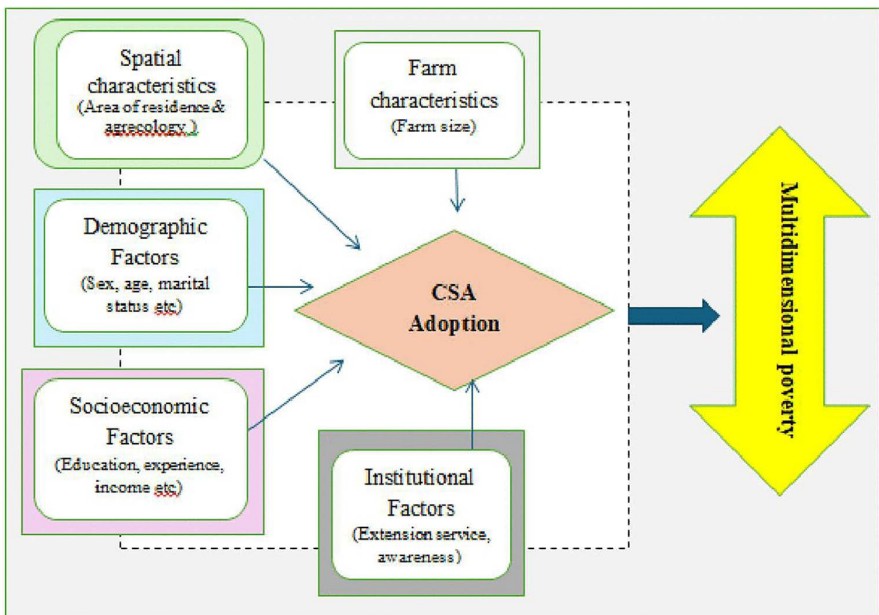

**Fig 1. Conceptual framework of the study.** Source: based on literature review (2024).

## Empirical review

**Climate smart agriculture and poverty reduction.** Empirical studies highlighted that CSA has role in enhancing agricultural productivity, resilience, and rural livelihoods [4,6,10] However, few studies have examined its effect on multidimensional poverty, particularly in consideration with agroecological and spatial variations. For instance, [4] found that Farmers who adopt CSA in the Ethiopian Rift Valley enjoyed higher food consumption scores and lower multidimensional poverty. Similarly, [11] Uses panel data from Ethiopia to analyse conservation agriculture techniques (minimum tillage, cereal-legume intercropping) finds that these practices significantly reduce the incidence and depth of poverty, especially in rainfall-stressed areas.

**Methodological approaches.** Previous studies employed different methodological approach to investigate the effects of CSA technologies adoption on food security/welfare/poverty. For instance, endogenous switch regression used by [4,11] propensity score matching (PSM) used by [12–14] Difference-in-Differences used by [15]. Nevertheless, each of them has advantages and limitations a growing number of studies acknowledge the endogeneity problem. Though, limited studies uses 2SLS (control endogeneity problem by using appropriate instrument variables) [16] and GSEM (appropriate for latent and mediation variables and accommodates binary, ordinal and count dependent variable) [17]. This study simultaneously employed GSEM, PSM and 2SLS analytical models to examine the impact of multiple CSA practices adoption on multidimensional poverty.

## Methodology of the study

### Description of the study area

This study was conducted in the Central Ethiopia Region, Silte Zone, which is in the South of Ethiopia and encompasses nine districts. The eco-region lies between 7° 49' 56" N latitude to 38° 16' 7" E longitude, with an elevation of 1967 meters. In the zone, there are two main livelihoods: Enset with mixed crop production in the highland and mixed crop production in the midland.

## Research design

This study employed quantitative research design following [18].

## Sampling techniques

A multistage sampling technique was employed. First, districts were purposively selected based on agroecological characteristics and relevance to CSA practices. Second, kebeles (the smallest administrative units in Ethiopia) were randomly selected. Finally, households were selected using systematic random sampling to ensure representativeness. The recruitment period was from 10, May 2024 up to 20, August 2024

## Types, sources, and methods of data collection

A structured questionnaire employed and primary data collected from selected respondents.

## Methods of data analysis

As shown in Fig 2, Sustainable livelihood framework (SLF) was adopted as one of the main theoretical underpinnings to lead the analysis of smallholder exposure and adaptation to climate change and its proceeding impact [7]. This framework is very helpful to comprehend the link among livelihood, risk (hazard, exposure and vulnerability for the consequence of climate change), their adaptation strategies and poverty reduction impact. The SLF consists of five sections; the vulnerability context; livelihood assets; policy, institutions and procedures; livelihood strategies; and livelihood outcomes. In this study, we refer to vulnerability or exposure as the climate change and its associated shock (erosion and pest infestation). It frequently impacts households in the study area [19,20].

## Empirical models specification

The MPI was constructed using indicators related to education, health, and living standards, livelihood and risk exposure. Each indicator was weighted following the standard multidimensional poverty measurement framework proposed by [21], which assigns weights across dimensions of deprivation and applies a poverty cutoff of 0.33 to identify multidimensionally poor households. Based on this framework, households were categorized as non-poor (MPI < 0.33), vulnerable (0.33 ≤ MPI < 0.50), poor (0.50 ≤ MPI < 0.75), and severely poor (MPI ≥ 0.75), following extensions commonly applied in

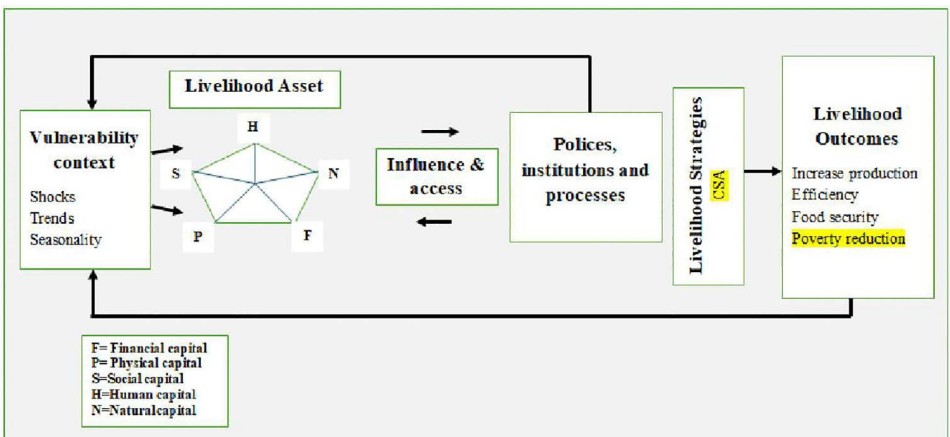

**Fig 2. Sustainable livelihood frameworks.** Source: Adopted from (Chambers R & Conway, 1992), with own modification.

empirical studies. This study employs three complementary econometric approaches to address different empirical challenges in analysing the relationship between CSA adoption and multidimensional poverty.

First, the Generalized Structural Equation Model (GSEM) is used to analyse the ordered structure of the Multidimensional Poverty Index (MPI) categories (non-poor, vulnerable, poor, and severely poor). GSEM allows the simultaneous estimation of multiple relationships and captures the structural links between CSA adoption and poverty outcomes. GSEM is particularly suitable for modelling latent structures and multiple interdependent relationships in socio-economic data [22]

Second, Propensity Score Matching (PSM) is applied to estimate the causal impact of CSA adoption on poverty outcomes by correcting for selection bias between adopters and non-adopters, based on the conditional independence assumption [23,24]

Third, the Two-Stage Least Squares (2SLS) model is used to address potential endogeneity between CSA adoption and multidimensional poverty outcomes by employing instrumental variables [25,26]. Previous studies adopted similar method [27–31]. MPI was used as order categories (non-poor, vulnerable, poor and severely-poor), for GSEM and as continuous variable for PSM and 2SLS.

Extension contact and climate change awareness are valid instruments because they are strongly associated with CSA adoption through information provision and risk perception mechanisms, as supported by diffusion and behavioural theories [32]. At the same time, they do not directly influence multidimensional poverty outcomes except through CSA adoption, satisfying the exclusion restriction under appropriate controls.

The exogenous variables are included in the model are sex, age, marital status, family size, farm size experience, education, income, area of residence and agroecology. Furthermore, endogenous test done to check endogenous problem, first-stage regression summary statistics that tells us usage of strong instrument and also over identifying restrictions test was done. The 2SLS model formula can be written as:

The First Stage: The endogenous variable, x, is regressed on a set of instrumental variables, z, and other exogenous variables, w, using ordinal least squares (OLS):

$$x_i = \pi_0 + \pi_1 z_{1i} + \dots \pi_k z_{ki} + \gamma_1 w_{1i} + \dots + \gamma_m w_{mi} + u_i \tag{1}$$

Where $x_i$ is the endogenous variables, z is the instrumental variable, w is the exogenous variables, u is the error term. The predicted values of x, denoted as $\hat{x}$, are obtained from the first-stage regression:

$$\hat{x}_i = \hat{\pi}_0 + \hat{\pi}_1 z_{1i} + \dots \hat{\pi}_k z_{ki} + \hat{\gamma}_1 w_{1i} + \dots + \hat{\gamma}_m w_{mi} \tag{2}$$

The second stage: the original equation is re-estimated using OLS, but the endogenous variable, x, is replaced with its predicted values. That is:

$$y_i = \beta_0 + \beta_1 \hat{x}_i + \varepsilon_i \tag{3}$$

The generalized structural equation model (GSEM) specified as follow:

$$\Pr\left(Y = \frac{y}{z}\right) = \Pr\left(Y^* < k_y - z\right) - \Pr\left(Y^* < k_{y-1} - z\right) \tag{4}$$

Where, $Y^*$ is the underlying stochastic component for $Y$. The distribution for $Y^*$ is determined by the link function. GSEM allows logit, probit, and cloglog for the ordinal family. The probit link assigns $Y^*$ the standard normal distribution that is synonymous with the probit link for Bernoulli outcomes. So, the latent (unobserved level of poverty) specified as:

$$Y^* = \beta_0 + \beta_1 CSA + \beta'_2 x + \varepsilon \tag{5}$$

Where, $Y^*$ are the unobserved categories of multidimensional poverty and $\varepsilon$ is error term. The links to observed categories of MPI area as follow:

$$Y = \begin{cases} 1 \ if \ Y^* \leq z_1 \\ 2 \ if \ z_1 < Y^* \leq z_2 \\ 3 \ if \ z_2 < Y^* \leq z_3 \\ \quad . \\ \quad . \\ \quad . \\ k \ if \ z_{k-1} < Y^* \end{cases}$$

(6)

Where, $z_1$, $z_2$, $z_3$…..$z_{k-1}$ are the cut-off points.

PSM was used to estimate the treatment effect of adopting different CSA practices, Y represented the outcomes (MPI score) dependent on a set of characteristics of farm households denoted as "j" and "i" respectively.

$$Y = \alpha + \tau d_j + \beta x_{ij} + \varepsilon$$

(7)

The average MPI level difference between the treatment (CSA practices) and control (non-users) groups is seen in Equation 8 ($\tau ATE$). Since unobservable factors may correlate with the use of CSA practices and omitted factors that affect MPI, it is not easy to accurately estimate the impacts. As a result, the Average Effect on the Treated (ATT) is recommended over the Average Treatment Effect (ATE) since the study aimed to evaluate the actual impact of CSA practices on adopters, not a hypothetical full adoption scenario. and making it particularly relevant for policy analysis and program evaluation.

$$\tau ATE = E\left[\frac{Y}{X}, d = 1\right] - E[\frac{Y}{X}, d = 0]$$

(8)

$Y_1$ and $Y_0$ in equation 9 represent the MPI scores of CSA practices user against non-users. $E[Y_0/d = 1]$ Would represent the counterfactual result for the treated groups if they did not use CSA practices

$$ATE = E\left[Y_1/d = 1\right] - E[Y_0/d = 1]$$

(9)

Subsequent matching process was conducted to compare farm households that adopted CSA practices with those that have not, but possess similar characteristics. The propensity score denoted as $p(x)$ and based on a set of characteristics x as outlined by [23], was utilized in this analysis:

$$p(x) = P[\left[P_\gamma[d = \frac{1}{x}\right] = E[d/x]$$

(10)

Generally there are three assumptions to utilize PSM. The first one is Conditional Independence Assumption (CIA), also referred to as the Un-confoundedness Assumption. This assumption necessitates that the outcome variables $Y_0$ exhibit independence from the treatment. Second, the presence of common support ensures an appropriate basis for comparison. The third, limit matches to users and non-users who share common support within the distribution of propensity scores [33].

As shown in Table 1, this study considers seven CSA practices as a package and analysed the effects of adoptions of CSA practices on MPI in the study area considering two agroecological zones.

**Table 1. Description of CSA practices.**

| CSA Practices | Adopters (%) | Non-Adopters (%) |
|---|---|---|
| High yielding verities (HYV) | 49% | 51% |
| Soil and water conservation (SWC) | 68% | 32% |
| Spacing | 56% | 44% |
| Crop rotation | 54% | 46% |
| Irrigation | 8% | 92% |
| Crop diversification | 39% | 61% |
| Integrated pest management (IPM) | 52% | 48% |

Source: Model result, 2024.

## Ethical statement

This study passed through Addis Ababa University's institutional review board. The IRB rigorously examined the proposal's content, its associated research tools, and the informed consent of the respondents. (Oral written participant consent since most respondents are illiterate farmers) and it witnessed and approved by the IRB and finally ethical clearance letter are issued ahead of the data collection. The approval no. of this ethical clearance is 075/04/2024.

## Results and discussions

### Descriptive results

As indicated in Table 2, Most of the households in the study area were middle-aged farmers. The implication I they are economically active and physically productive stage. This implies greater ability to engage in labor-intensive farming activities and higher potential to manage multiple farm operations efficiently. The average farm size is 1.99 hectare in the study area, which is greater than the average farm size (0.96 hectares) estimated by [34] for high-potential areas, and the national average farm size (0.9 hectares) of smallholders in Ethiopia.

### Results of econometric models

Hypothesis: Adopting CSA practices decrease MPI level of farm households.

To put the hypothesis to the test and calculate the effect of adopting CSA practices, MPI were used as outcome variable. Different CSA practices adopters were categorized as the treatment group and non-adopters as the control group. Variables that have an equal impact on the control and treatment groups were chosen to calculate the outcome variables with the assumption that the inclusion of unimportant variables and the exclusion of important variables greatly bias the results [24,35]

As indicated in Table 3, testing the Variance Inflation Factor (VIF) and in Table 4, Contingency Coefficient (C) for the continuous and dummy/categorical variables was done to reduce the impact of multicollinearity. The results showed that the mean values of 1.09 and that the dummy variables had C close to zero. Thus, it is determined that the model is free from multicollinearity, giving us the confidence to move forward with our regression.

### Estimation of propensity score matching

As indicated in Table 5, Ps-test was used to create a balance between the covariates of the two groups. Because of this, the bias percentage that ranged between 0.3 and 89.1 before matching has decreased to 0.1 to 16.4 after matching. This shows that the bias percentage has kept well below the 20% critical threshold cut-off point, indicating a minimized

**Table 2. Summary of explanatory variables used in the model.**

| Variables | Description | Non-Adopters | Adopter | Mean difference | t-value |
|---|---|---|---|---|---|
| Sex | Sex of household head (1 if Male; 0 otherwise) | 0.18 | 0.77 | −0.77 | −3.2*** |
| Age | Age of the Household head | 56 | 44 | 12 | 2.0** |
| Marital status | Marital status of HHH | 1 | 1.94 | −0.94 | −4.3*** |
| Family size | Household size (in adult equivalent) | 58 | 72 | −14 | −0.88 |
| Farm size | Total landholding size (in hectare) | 1 | 3.58 | −2.58 | −2.48*** |
| Education | HHH level of schooling | 2 | 3.94 | −3.97 | −2.02** |
| Experience | HHH farm experience | 30 | 16.1 | 14 | 2.46*** |
| Total income | HH total income (in birr) | 6500 | 17640 | −11140 | −2.30** |
| Extension | Frequency of extension contacts | 5 | 3.23 | −3.23 | −3.04*** |
| Awareness | Awareness of HH about climate change | 0.41 | 0.63 | −0.63 | −2.27** |
| Area of residence | Area of residence of households (1 if Rural, 0 Peri-urban) | 0.33 | 0.49 | 0.16 | −0.55 |
| Agroecology | Agroecology of households where they located (1 if Highland, 0 midland) | 0.33 | 0.56 | −0.56 | −0.79 |

Source: Model result, 2024.

**Table 3. Result of VIF.**

| Variable | VIF | 1/VIF |
|---|---|---|
| Total farm size | 2.65 | 0.377517 |
| Total income | 2.63 | 0.379946 |
| Labor (ADeq) | 1.18 | 0.845274 |
| Livestock (TLU) | 1.18 | 0.848100 |
| Education | 1.02 | 0.977040 |
| Age | 1.02 | 0.981755 |
| Mean VIF | 1.61 | |

Source: Model result, 202.

**Table 4. C test for discrete variables.**

| | Area | Agroecology | Sex | Marital status |
|---|---|---|---|---|
| Area | 1.000 | | | |
| Agroecology | −0.0201 | 1.000 | | |
| Sex | 0.0579 | 0.0376 | 1.000 | |
| Marital status | 0.0244 | −0.0288 | −0.0842 | 1.000 |

Source: Model result, 2024

imbalance between the treatment and control samples [23]. As a result, the covariate balance between the treatment and control samples is greatly enhanced, which can be employed in subsequent estimation processes.

As illustrated in Table 6, from kernel matching methods, kernel with bandwidth 0.01 was chosen, by using the values of Ps $R^2$ and LR chi$^2$ as indicators for completing the balancing criteria. The assumption that both groups have a similar distribution in covariates after matching is confirmed by the relatively low Ps $R^2$ 0.011, 0.008, 0.003, 0.01, 0.007, 0.0160 and 0.005 for HYV, SWC, spacing, crop rotation, irrigation, crop diversification and IPM respectively. Furthermore, the

**Table 5. Propensity score and covariate matching.**

| Treatment variable High yielding varieties (HYV) | | | | | | |
|---|---|---|---|---|---|---|
| Variables | Unmatched/ Matched | Mean | | % bias | % reduction/bias/ | P>/t/ |
| | | Treated | Controlled | | | |
| Area | U | 0.64706 | 0.24171 | 89.1 | | 0.000 |
| | M | 0.62147 | 0.59063 | 6.8 | 92.4 | 0.554 |
| Agroecology | U | 0.57843 | 0.53555 | 8.6 | | 0.381 |
| | M | 0.58192 | 0.57385 | 1.6 | 81.2 | 0.878 |
| Sex | U | 0.67157 | 0.55924 | 23.2 | | 0.019 |
| | M | 0.64407 | 0.70354 | −12.3 | 47.1 | 0.234 |
| Marital status | U | 1.4706 | 1.4502 | 3.4 | | 0.727 |
| | M | 1.4746 | 1.415 | 10.1 | −192.9 | 0.342 |
| Education | U | 3.6029 | 3.4265 | 5.4 | | 0.581 |
| | M | 3.5932 | 3.799 | −6.3 | −16.6 | 0.557 |
| Age | U | 43.275 | 44.673 | −14.2 | | 0.149 |
| | M | 43.446 | 42.033 | 14.3 | −1.1 | 0.174 |
| Age2 | U | 1965.6 | 2096.2 | −14.7 | | 0.136 |
| | M | 1982.7 | 1861.6 | 13.6 | 7.2 | 0.192 |
| Labor (ad.eq) | U | 8.238 | 8.5374 | −9.3 | | 0.342 |
| | M | 8.1292 | 8.4671 | −10.6 | −12.9 | 0.308 |
| Total farm size | U | 2.5441 | 1.3488 | 71.4 | | 0.000 |
| | M | 2.2486 | 2.3919 | −8.6 | 88.0 | 0.94 |
| Total income | U | 11311 | 7708.5 | 67.1 | | 0.000 |
| | M | 10119 | 10301 | −3.4 | 94.9 | 0.775 |
| **Treatment variable Soil and water conservation (SWC)** | | | | | | |
| Variables | Unmatched/ Matched | Mean | | % bias | % reduction/bias/ | P>/t/ |
| | | Treated | Controlled | | | |
| Area | U | 0.52482 | 0.26316 | 55.4 | | 0.000 |
| | M | 0.46025 | 0.48141 | −4.5 | 91.9 | 0.644 |
| Agroecology | U | 0.57447 | 0.5188 | 11.2 | | 0.288 |
| | M | 0.57322 | 0.54601 | 5.5 | 51.1 | 0.550 |
| Sex | U | 65248 | 0.53383 | 24.3 | | 0.020 |
| | M | 0.61506 | 0.60055 | 3.0 | 87.8 | 0.746 |
| Marital status | U | 1.4326 | 1.5188 | −14.4 | | 0.167 |
| | M | 1.4644 | 1.5629 | −16.4 | −14.3 | 0.097 |
| Education | U | 3.7234 | 3.0677 | 20.4 | | 0.055 |
| | M | 3.5397 | 3.6949 | −4.8 | 76.3 | 0.613 |
| Age | U | 43.603 | 44.797 | −12.0 | | 0.251 |
| | M | 43.891 | 45.059 | −11.8 | 2.2 | 0.197 |
| Age2 | U | 1996.4 | 2107.4 | −12.3 | | 0.237 |
| | M | 2025.9 | 2125.7 | −11.1 | 10.0 | 0.221 |
| Labor (ad.eq) | U | 8.2832 | 8.6172 | −10.4 | | 0.322 |
| | M | 8.3145 | 7.9082 | 12.6 | −21.6 | 0.176 |
| Total farm size | U | 2.2583 | 1.2538 | 63.7 | | 0.000 |
| | M | 1.8299 | 1.8334 | −0.2 | 99.7 | 0.981 |
| Total income | U | 10463 | 7394.7 | 63.9 | | 0.000 |
| | M | 8788.7 | 8840.6 | −1.1 | 98.3 | 0.903 |

*(Continued)*

**Table 5.** (Continued)

| Treatment variable High yielding varieties (HYV) | | | | | | |
|---|---|---|---|---|---|---|
| **Variables** | **Unmatched/ Matched** | **Mean** | | **% bias** | **% reduction/bias/** | **P>/t/** |
| | | **Treated** | **Controlled** | | | |
| **Treatment variable Spacing** | | | | | | |
| **Variables** | **Unmatched/ Matched** | **Mean** | | **% bias** | **% reduction/bias/** | **P>/t/** |
| | | **Treated** | **Controlled** | | | |
| Area | U | 0.54701 | 0.30387 | 50.6 | | 0.000 |
| | M | 0.48718 | 0.51102 | −5.0 | 90.2 | 0.639 |
| Agroecology | U | 0.55983 | 0.55249 | 1.5 | | 0.882 |
| | M | 0.55385 | 0.5751 | −4.3 | −189.4 | 0.673 |
| Sex | U | 0.66239 | 0.55249 | 22.6 | | 0.023 |
| | M | 0.64615 | 0.63054 | 3.2 | 85.8 | 0.749 |
| Marital status | U | 1.4915 | 1.4199 | 12.2 | | 0.222 |
| | M | 1.4923 | 1.4744 | 3.0 | 75.0 | 0.772 |
| Education | U | 3.641 | 3.3481 | 9.0 | | 0.363 |
| | M | 3.5897 | 3.4327 | 4.8 | 46.4 | 0.626 |
| Age | U | 43.568 | 44.525 | −9.7 | | 0.328 |
| | M | 43.887 | 44.571 | −6.9 | 28.5 | 0.495 |
| Age2 | U | 1993.7 | 2081.5 | −9.8 | | 0.321 |
| | M | 2024.6 | 2082.8 | −6.5 | 33.7 | 0.524 |
| Labor (ad.eq) | U | 8.1113 | 8.7508 | −20.1 | | 0.043 |
| | M | 8.123 | 8.0064 | 3.7 | 81.8 | 0.722 |
| Total farm size | U | 2.3509 | 1.4006 | 56.6 | | 0.000 |
| | M | 2.0082 | 1.9765 | 1.9 | 96.7 | 0.858 |
| Total income | U | 10767 | 7814.9 | 56.3 | | 0.000 |
| | M | 9033.3 | 8927.4 | 2.0 | 96.4 | 0.829 |
| **Treatment variable crop rotation** | | | | | | |
| **Variables** | **Unmatched/ Matched** | **Mean** | | **% bias** | **% reduction/bias/** | **P>/t/** |
| | | **Treated** | **Controlled** | | | |
| Area | U | 0.59375 | 0.26178 | 71.1 | | 0.000 |
| | M | 0.565 | 0.56032 | 1.0 | 98.6 | 0.925 |
| Agroecology | U | 0.57589 | 0.53403 | 8.4 | | 0.393 |
| | M | 0.565 | 0.60839 | −8.7 | −3.7 | 0.379 |
| Sex | U | 0.66518 | 0.55497 | 22.7 | | 0.021 |
| | M | 0.65 | 0.65266 | −0.5 | 97.6 | 0.956 |
| Marital status | U | 1.4911 | 1.4241 | 11.3 | | 0.251 |
| | M | 1.475 | 1.4417 | 5.6 | 50.3 | 0.582 |
| Education | U | 3.5089 | 3.5183 | −0.3 | | 0.977 |
| | M | 3.485 | 3.2765 | 6.4 | −2118.8 | 0.513 |
| Age | U | 43.589 | 44.45 | −8.7 | | 0.377 |
| | M | 43.66 | 43.097 | 5.7 | 34.6 | 0.579 |
| Age2 | U | 1999.2 | 2070.4 | −8.0 | | 0.418 |
| | M | 2003 | 1964.7 | 4.3 | 46.3 | 0.674 |
| Labor (ad.eq) | U | 8.2665 | 8.5354 | −8.4 | | 0.395 |
| | M | 8.264 | 7.8022 | 14.4 | −71.8 | 0.166 |
| Total farm size | U | 2.4089 | 1.3822 | 61.0 | | 0.000 |
| | M | 2.2455 | 2.1414 | 6.2 | 89.9 | 0.590 |

*(Continued)*

**Table 5.** (Continued)

| Treatment variable High yielding varieties (HYV) | | | | | | |
|---|---|---|---|---|---|---|
| **Variables** | **Unmatched/ Matched** | **Mean** | | **% bias** | **% reduction/bias/** | **P>/t/** |
| | | **Treated** | **Controlled** | | | |
| Total income | U | 10848 | 7874.3 | 56.0 | | 0.000 |
| | M | 9800 | 9324.3 | 9.0 | 84.0 | 0.388 |
| **Treatment variable Irrigation** | | | | | | |
| **Variables** | **Unmatched/ Matched** | **Mean** | | **% bias** | **% reduction/bias/** | **P>/t/** |
| | | **Treated** | **Controlled** | | | |
| Area | U | 0.54545 | 0.43194 | 22.7 | | 0.209 |
| | M | 0.53125 | 0.52001 | 2.2 | 90.1 | 0.930 |
| Agroecology | U | 0.60606 | 0.55236 | 10.8 | | 0.552 |
| | M | 0.59375 | 0.59437 | −0.1 | 98.8 | 0.996 |
| Sex | U | 0.69697 | 0.60733 | 18.8 | | 0.311 |
| | M | 0.6875 | 0.64475 | 8.9 | 52.3 | 0.722 |
| Marital status | U | 1.6061 | 1.4476 | 23.7 | | 0.140 |
| | M | 1.5938 | 1.5552 | 5.8 | 75.6 | 0.826 |
| Education | U | 4.7273 | 3.4084 | 39.9 | | 0.025 |
| | M | 4.5625 | 4.4981 | 1.9 | 95.1 | 0.940 |
| Age | U | 39.848 | 44.343 | −50.1 | | 0.012 |
| | M | 39.906 | 40.428 | −5.8 | 88.4 | 0.801 |
| Age2 | U | 1647.6 | 2065.2 | −52.5 | | 0.010 |
| | M | 1654 | 1704.1 | −6.3 | 88.0 | 0.775 |
| Labor (ad.eq) | U | 8.7589 | 8.3584 | 12.4 | | 0.491 |
| | M | 8.6576 | 8.7245 | −2.1 | 83.3 | 0.934 |
| Total farm size | U | 2.3485 | 1.9008 | 27.8 | | 0.164 |
| | M | 2.3281 | 2.1926 | 8.4 | 69.7 | 0.753 |
| Total income | U | 10545 | 9387.4 | 19.9 | | 0.257 |
| | M | 10094 | 10188 | −1.6 | 91.9 | 0.950 |
| **Treatment variable crop diversification** | | | | | | |
| **Variables** | **Unmatched/ Matched** | **Mean** | | **% bias** | **% reduction/bias/** | **P>/t/** |
| | | **Treated** | **Controlled** | | | |
| Area | U | 0.64634 | 0.30677 | 72.1 | | 0.000 |
| | M | 0.61905 | 0.57817 | 8.7 | 88.0 | 0.476 |
| Agroecology | U | 0.56707 | 0.5498 | 3.5 | | 0.730 |
| | M | 0.57143 | 0.53119 | 8.1 | −133.0 | 0.490 |
| Sex | U | 0.68902 | 0.56574 | 25.6 | | 0.012 |
| | M | 0.67347 | 0.69803 | −5.1 | 80.1 | 0.652 |
| Marital status | U | 1.4939 | 1.4382 | 9.3 | | 0.350 |
| | M | 1.4762 | 1.5397 | −10.6 | −14.0 | 0.383 |
| Education | U | 3.3293 | 3.6335 | −9.4 | | 0.352 |
| | M | 3.3129 | 2.9377 | 11.6 | −23.3 | 0.306 |
| Age | U | 43.665 | 44.195 | −5.3 | | 0.593 |
| | M | 43.585 | 44.244 | −6.6 | −24.1 | 0.581 |
| Age2 | U | 2009.2 | 2046.9 | −4.2 | | 0.674 |
| | M | 2004.8 | 2059.5 | −6.1 | −45.2 | 0.610 |

*(Continued)*

**Table 5.** (Continued)

| Treatment variable High yielding varieties (HYV) | | | | | | |
|---|---|---|---|---|---|---|
| **Variables** | **Unmatched/ Matched** | **Mean** | | **% bias** | **% reduction/bias/** | **P>/t/** |
| | | **Treated** | **Controlled** | | | |
| Labor (ad.eq) | U | 8.4773 | 8.3334 | 4.5 | | 0.655 |
| | M | 8.535 | 8.1523 | 12.0 | −165.8 | 0.315 |
| Total farm size | U | 2.5732 | 1.5203 | 60.4 | | 0.000 |
| | M | 2.2517 | 2.4399 | −10.8 | 82.1 | 0.391 |
| Total income | U | 11561 | 8119.5 | 59.8 | | 0.000 |
| | M | 10150 | 10064 | 1.5 | 97.5 | 0.900 |
| **Treatment variable Integrated pest management (IPM)** | | | | | | |
| **Variables** | **Unmatched/ Matched** | **Mean** | | **% bias** | **% reduction/bias/** | **P>/t/** |
| | | **Treated** | **Controlled** | | | |
| Area | U | 0.55556 | 0.31658 | 49.5 | | 0.000 |
| | M | 0.52632 | 0.51931 | 1.5 | 97.1 | 0.892 |
| Agroecology | U | 0.56019 | 0.55276 | 1.5 | | 0.880 |
| | M | 0.55263 | 0.58022 | −5.5 | −271.7 | 0.589 |
| Sex | U | 0.67593 | 0.54774 | 26.5 | | 0.007 |
| | M | 0.64211 | 0.6749 | −6.8 | 74.4 | 0.502 |
| Marital status | U | 1.4537 | 1.4673 | −2.3 | | 0.815 |
| | M | 1.4474 | 1.4355 | 2.0 | 13.0 | 0.843 |
| Education | U | 3.8472 | 3.1508 | 21.6 | | 0.029 |
| | M | 3.7842 | 3.9401 | −4.8 | 77.6 | 0.653 |
| Age | U | 43.519 | 44.492 | −9.9 | | 0.316 |
| | M | 43.716 | 44.038 | −3.3 | 66.9 | 0.756 |
| Age2 | U | 1993.9 | 2073.3 | −8.9 | | 0.366 |
| | M | 2010.2 | 2043.7 | −3.8 | 57.8 | 0.721 |
| Labor (ad.eq) | U | 8.3607 | 8.4223 | −1.9 | | 0.845 |
| | M | 8.3785 | 8.4783 | −3.1 | −62.1 | 0.759 |
| Total farm size | U | 2.4282 | 1.4025 | 60.9 | | 0.000 |
| | M | 2.0579 | 2.0459 | 0.7 | 98.8 | 0.945 |
| Total income | U | 11023 | 7804 | 60.5 | | 0.000 |
| | M | 9792.1 | 9387.7 | 7.6 | 87.4 | 0.471 |

Source: Model result, 2024.

insignificant LR ch2 5.26, 5.52, 1.82, 5.48, 0.61, 6.54, and 2.82 after matching for HYV, SWC, spacing, crop rotation, irrigation, crop diversification and IPM respectively. It is supported by [36–39].

As indicated in Table 7, According to the Minima and Maxima criterion, observations with a propensity score smaller than and larger than the opposing group were eliminated while determining the common support region [24] As a result, the range of common support is between (0.138 and 0.970), (0.325 and 0.995), (0.261 and 0.949), (0.207 and 0.956), (0.092 and 0.913), (0.0008 and 0.456) and (0.267 and 0.953) for HYV, SWC, spacing, crop rotation, crop diversification, irrigation and IPM respectively and any households outside of this range were excluded from the matching process. The result is confirmed by [36–39].

**Table 6. Comparison of the matching estimators.**

| Performance criteria | HYV | | | Performance criteria | SWC | | |
|---|---|---|---|---|---|---|---|
| | BW (0.05) | BW (0.01) | BW (0.1) | | BW (0.05) | BW (0.01) | BW (0.1) |
| PsR$^2$ | 0.016 | **0.011** | 0.016 | PsR$^2$ | 0.037 | **0.008** | 0.031 |
| LR chi$^2$ | 8.850 | **5.26** | 8.85 | LR chi$^2$ | 29.27 | **5.52** | 24.59 |
| p>chi$^2$ | 0.547 | **0.873** | 0.547 | p>chi$^2$ | 0.001 | **0.854** | 0.006 |
| MeanBias | 6.0 | **8.8** | 6.0 | MeanBias | 14.5 | **7.1** | 11.5 |
| MedBias | 5.9 | **9.3** | 5.9 | MedBias | 12.6 | **5.1** | 12.3 |
| B | 29.3* | **24.4** | 29.3* | B | 46.1* | **21.5** | 41.8* |
| R | 2.28* | **1.31** | 2.28* | R | 1.10 | **0.83** | 2.10* |
| %Var | 14 | **0** | 14 | %Var | 14 | **14** | 29 |
| **Performance criteria** | **HYV** | | | **Performance criteria** | **SWC** | | |
| | BW (0.05) | BW (0.01) | BW (0.1) | | BW (0.05) | BW (0.01) | BW (0.1) |
| PsR$^2$ | 0.017 | **0.003** | 0.012 | PsR$^2$ | 0.019 | **0.01** | 0.014 |
| LR chi$^2$ | 11.29 | **1.82** | 7.77 | LR chi$^2$ | 11.64 | **5.48** | 8.52 |
| p>chi$^2$ | 0.335 | **0.998** | 0.652 | p>chi$^2$ | 0.310 | **0.857** | 0.578 |
| MeanBias | 7.7 | **4.1** | 6.0 | MeanBias | 5.6 | **6.2** | 4.2 |
| MedBias | 5.5 | **4.0** | 5.1 | MedBias | 5.1 | **5.9** | 2.3 |
| B | 31.1* | **13.7** | 25.6* | B | 31.8* | **23.4** | 27.2* |
| R | 1.37 | **1.03** | 1.94 | R | 2.89* | **1.06** | 3.76* |
| %Var | 14 | **0** | 14 | %Var | 14 | **0** | 14 |
| **Performance criteria** | **Irrigation** | | | **Performance criteria** | **Crop diversification** | | |
| | BW (0.05) | BW (0.01) | BW (0.1) | | BW (0.05) | BW (0.01) | BW (0.1) |
| PsR$^2$ | **0.007** | 0.008 | 0.023 | PsR$^2$ | 0.018 | **0.016** | 0.014 |
| LR chi$^2$ | **0.61** | 0.69 | 2.08 | LR chi$^2$ | 8.16 | **6.54** | 6.24 |
| p>chi$^2$ | **1.00** | 1.00 | 0.996 | p>chi$^2$ | 0.613 | **0.768** | 0.795 |
| MeanBias | **4.3** | 4.2 | 9.3 | MeanBias | 7.4 | **8.1** | 6.5 |
| MedBias | **4.0** | 2.6 | 10.3 | MedBias | 6.7 | **8.4** | 6.8 |
| B | **19.3** | 20.5 | 35.6* | B | 31.6* | **29.8*** | 27.3* |
| R | **0.87** | 0.40* | 0.67 | R | 2.40* | **1.28** | 3.00* |
| %Var | **0** | 0 | 0 | %Var | 14 | **14** | 14 |
| **Performance criteria** | **IPM** | | | | | | |
| | BW (0.05) | BW (0.01) | BW (0.1) | | | | |
| PsR$^2$ | 0.006 | **0.005** | 0.007 | | | | |
| LR chi$^2$ | 3.52 | **2.82** | 4.38 | | | | |
| p>chi$^2$ | 0.967 | **0.985** | 0.929 | | | | |
| MeanBias | 3.9 | **3.9** | 4.6 | | | | |
| MedBias | 3.5 | **3.5** | 3.8 | | | | |
| B | 18.0 | **17.1** | 20.0 | | | | |
| R | 2.00* | **1.73** | 2.54* | | | | |
| %Var | 0 | **14** | 14 | | | | |

Source: Model result, 2024.

**Table 7. Distribution of estimated propensity scores.**

| Variable | Group | Obs. | Mean | SD | Min | Max |
|---|---|---|---|---|---|---|
| Pscore (HYV) | Total | 415 | 0.4923984 | 0.2428627 | 0.1389612 | 0.9700462 |
| | Treated | 204 | 0.6127755 | 0.238342 | 0.1389612 | 0.9700462 |
| | Controlled | 211 | 0.3760147 | 0.1835203 | 0.1471128 | 0.9646568 |
| Pscore (SWC) | Total | 415 | 0.6798698 | 0.1686818 | 0.3198355 | 0.9957257 |
| | Treated | 282 | 0.7210681 | 0.1696418 | 0.3259327 | 0.9957257 |
| | Controlled | 133 | 0.5925169 | 0.1290865 | 0.3198355 | 0.9539493 |
| Pscore (Spacing) | Total | 415 | 0.5636643 | 0.1744424 | 0.2562972 | 0.9493206 |
| | Treated | 282 | 0.6028278 | 0.1788728 | 0.2611426 | 0.9493206 |
| | Controlled | 133 | 0.4806257 | 0.1307828 | 0.2562972 | 0.9473756 |
| Pscore (Crop rotation) | Total | 415 | 0.5386952 | 0.2021016 | 0.207716 | 0.9566356 |
| | Treated | 282 | 0.5825777 | 0.2050881 | 0.207716 | 0.9566356 |
| | Controlled | 133 | 0.445651 | 0.1604026 | 0.2169875 | 0.9387753 |
| Pscore (Crop diversification) | Total | 415 | 0.3942165 | 0.2086554 | 0.0927305 | 0.9136933 |
| | Treated | 282 | 0.4359596 | 0.2182523 | 0.0927305 | 0.9136933 |
| | Controlled | 133 | 0.3057086 | 0.1533941 | 0.0960882 | 0.9046335 |
| Pscore (Irrigation) | Total | 415 | 0.0795706 | 0.0681301 | 0.0008883 | 0.4560932 |
| | Treated | 282 | 0.0870917 | 0.0734612 | 0.0008883 | 0.4560932 |
| | Controlled | 133 | 0.0636237 | 0.0518931 | 0.0017508 | 0.2767974 |
| Pscore (IPM) | Total | 415 | 0.5193335 | 0.1801154 | 0.2607841 | 0.9531466 |
| | Treated | 282 | 0.5612729 | 0.1893624 | 0.2679787 | 0.9531466 |
| | Controlled | 133 | 0.4304094 | 0.1170007 | 0.2607841 | 0.9034773 |

Source: Model result, 2024.

Fig 3 depicts the propensity score of the distribution density of the treatment and control groups graphically. This indicates that the balance has been successfully attained because the p-score is fairly distributed between the treatment and control groups.

### Estimation of treatment effect on the treated

Table 8 shows the results of the impacts of the use of HYV, SWC, Spacing, crop rotation, Irrigation, crop diversification and IPM on the outcome variables of multidimensional poverty. According to the Kernel estimate, farm who adopted CSA practices significantly minimize MPI level by 0.20, 0.13, 0.12, 0.17, 0.04, 0.04 and 0.17 for HYV, SWC, spacing, crop rotation, irrigation, crop diversification and IPM adopters compared with non-adopters respectively. This finding supported by the findings of [36–39].

### Sensitivity analysis

As indicated in Table 9, Sensitivity analysis is conducted to evaluate how reliable or consistent the PSM results are [24,40]. Based on this, the robustness of hidden bias among the two outcome variables of the study is shown in Table 9. The Rosenbaum sensitivity analysis indicates that the lowest critical value of Γ at which the treatment effect becomes statistically insignificant is 8.8 (at the 95% confidence level). This implies that, due to unobserved heterogeneity, two otherwise similar households could differ in their odds of adopting climate-smart agriculture practices by a factor of up to 8.8 and still maintain the estimated treatment effect. This result has also indicated that for the MPI estimated at various levels

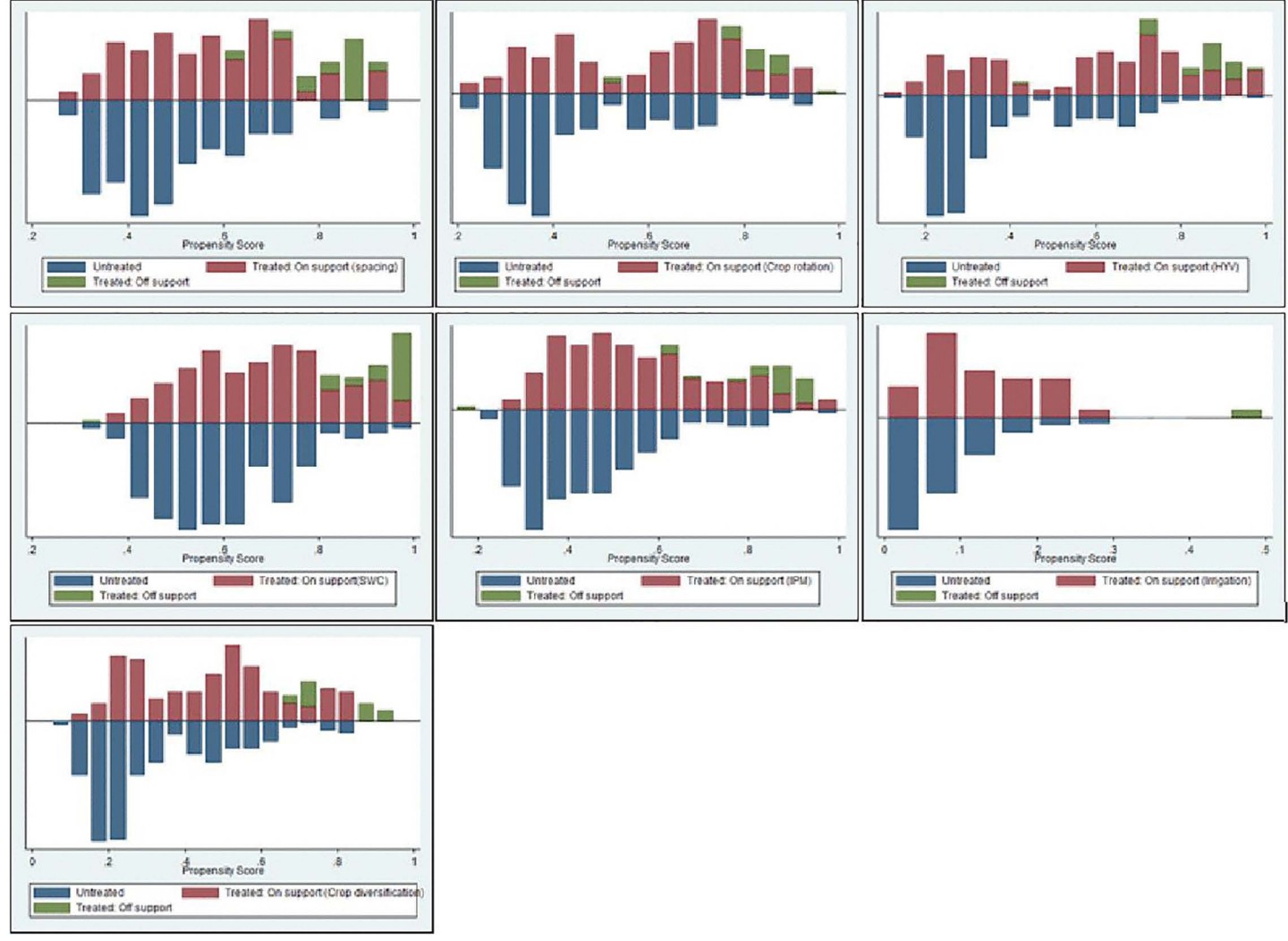

**Fig 3. Distribution of propensity score generated from Kernel Matching.** Source: Model result, 2024.

of critical values $e^{\gamma}$, the p-critical values are significant, informing that important covariates that affected the outcomes have been taken into account. Overall, the hidden bias magnitude confirms the hypothesis that the findings of significant differences in MPI level between users and non-users are insensitive to hidden biases; as a result, the positive impacts found are attributed to the use of different CSA practices.

## Estimation of two stage least square (2SLS)

As Table 10 show the results of the effect of the use of HYV, SWC, Spacing, crop rotation, Irrigation, crop diversification and IPM on the outcome variables of multidimensional poverty. Farmer who adopted CSA practices significantly minimize MPI level by 15%, 25%, 25%, 15%, 18%, 12% and 45% for HYV, SWC, spacing, crop rotation, irrigation, crop diversification and IPM adopters compared with non-adopters respectively. This finding supported by the findings of [36–39].

**Table 8. CSA practices adoption result of average treatment effect on the treated (ATT).**

| CSA practices | DV | Sample | Treated | Controlled | Difference | S.E | T-stat |
|---|---|---|---|---|---|---|---|
| HYV | MPI score | Unmatched | 0.585457514 | 0.836334906 | −0.250877392 | 0.018130887 | −13.84 |
| | | ATT | 0.585457514 | 0.785620911 | −0.200163397 | 0.022495792 | −8.90 |
| SWC | MPI score | Unmatched | 0.655555553 | 0.834837085 | −0.179281532 | 0.021779364 | −8.23 |
| | | ATT | 0.655555553 | 0.783569734 | −0.128014181 | 0.025934461 | −4.94 |
| Spacing | MPI score | Unmatched | 0.638319086 | 0.80957642 | −0.171257333 | 0.020443784 | −8.38 |
| | | ATT | 0.638319086 | 0.760968656 | −0.12264957 | 0.023294732 | −5.27 |
| Crop rotation | MPI score | Unmatched | 0.608779759 | 0.835253048 | −0.226473288 | 0.018969962 | −11.94 |
| | | ATT | 0.608779759 | 0.777380948 | −0.168601188 | 0.023756353 | −7.10 |
| Irrigation | MPI score | Unmatched | 0.61313131 | 0.721640484 | −0.108509174 | 0.040178637 | −2.70 |
| | | ATT | 0.61313131 | 0.657575756 | −0.044444446 | 0.06483768 | −0.69 |
| Crop diversification | MPI score | Unmatched | 0.5550813 | 0.816201853 | −0.261120553 | 0.018384886 | −14.20 |
| | | ATT | 0.568253967 | 0.773330955 | −0.205076988 | 0.023351161 | −8.78 |
| IPM | MPI score | Unmatched | 0.615895059 | 0.818425454 | −0.202530395 | 0.019557042 | −10.36 |
| | | ATT | 0.615895059 | 0.790586414 | −0.174691354 | 0.022618323 | −7.72 |

Source: Model result, 2024.

## Estimation of generalized structural equation model (GSEM)

As indicated in Table 11, The GSEM is a key tool for causal analysis, although it offers tests to assess goodness of fit. The Akaike information criterion (AIC) and Bayesian information criterion (BIC) are used to measure the performance of the model proposed in this study. Table 4 shows the results of the model, AIC and BIC, control variables and different climate smart agriculture practices included in the model.

As shown in Fig 4, Based on GSEM model, households who adopted improved variety, SWC, spacing, crop rotation, crop diversification and IPM decrease the probability of being in a higher MPI category by 72.8%, 60%, 65%, 65%, 79% and 53% compared to non-adopters respectively. The result is confirmed by the study of [36–39].

The results of some control variables indicated a significant effect on MPI, supporting previous works that found these variables to be determinants to MPI. For instance, The coefficient of income indicates that a one-unit increase in household income is associated with a corresponding marginal reduction in multidimensional poverty, holding other factors constant. Living in a peri-urban area decreases the probability of being in a higher MPI category by 47% compared to rural farmers. Furthermore, marital status, age, and labor decrease the probability of being in higher MPI category by 26.7%, 12% and 0.7% respectively. [27,28,41] found the same result.

## Conclusion and recommendations

### Conclusion

This study investigated the pathways through which Climate-Smart Agriculture (CSA) adoption affects multidimensional poverty among rural and peri-urban households, using Generalized Structural Equation Model (GSEM), PSM and 2SLS. By treating both CSA practices separately as dummy variables and multidimensional poverty as continuous outcome variable.

The result from the PSM showed that the adoption of at least a single CSA practice drops the level of MPI at a range of 0.04–0.20. The findings reveal that CSA adoptions are significantly associated with lower levels of multidimensional poverty. Similarly, the results from GSEM that uses MPI as categorical outcome variable found low probability of CSA practices adopters to be in higher MPI category compared with non-adopters. Similarly the finding from 2SLS shows multidimensional poverty effects of different CSA practices adoption.

**Table 9. Rosenbaum bounds sensitivity analysis.**

| *Gamma (hidden bias) | Significance level | | Hodges-Lehmann point estimate | | Confidence interval (95%) | |
|---|---|---|---|---|---|---|
| | Upper bound (sig+) | Lower bound sig- | Upper bound (t-hat+) | Lower bound (t-hat-) | Upper bound (CI+) | Lower bound (CI-) |
| **Outcome variables MPI effect of HYV** | | | | | | |
| 1 | 0 | 0 | −0.15 | −0.15 | −0.216667 | −0.116667 |
| 2 | 0 | 1.6e-11 | −0.283333 | −0.083333 | −0.316667 | −0.066667 |
| **6.6** | **0** | **0.025777** | **−0.4** | **−0.016666** | **−0.45** | **3.8e-07** |
| **Outcome variables MPI effect of SWC** | | | | | | |
| 1 | 0 | 0 | −0.1 | −0.1 | −0.116667 | −0.066667 |
| 2 | 0 | 0.000032 | −0.166667 | −0.05 | −0.216667 | −0.016667 |
| **2.9** | **0** | **0.036286** | **−0.233333** | **−0.016667** | **−0.266667** | **4.9e-07** |
| **Outcome variables MPI effect of Spacing** | | | | | | |
| 1.5 | 0 | 0.000401 | −0.133334 | −0.033333 | −0.2 | −0.016666 |
| 1.6 | 0 | 0.001514 | −0.15 | −0.033333 | −0.216667 | −0.016666 |
| **2** | **0** | **0.049856** | **−0.2** | **−0.016666** | **−0.25** | **4.7e-07** |
| **Outcome variables MPI effect of crop rotation** | | | | | | |
| 1 | 0 | 0 | −0.133334 | −0.133334 | −0.166667 | −0.1 |
| 2 | 0 | 5.0e-09 | −0.233334 | −0.083333 | −0.283333 | −0.05 |
| **4.5** | **0** | **0.025583** | **−0.333334** | **−0.033333** | **−0.383334** | **3.2e-07** |
| **Outcome variables MPI effect of Irrigation** | | | | | | |
| 1.9 | 0.037561 | 0.930385 | −0.116666 | 0.066666 | −0.35 | 0.233333 |
| 2 | 0.027673 | 0.946584 | −0.116667 | 0.083333 | −0.35 | 0.233333 |
| **4.4** | **0.000012** | **0.999943** | **−0.316667** | **0.2** | **−0.616666** | **0.4** |
| **Outcome variables MPI effect Crop diversification** | | | | | | |
| 1 | 0 | 0 | −0.183334 | −0.183334 | −0.233333 | −0.133333 |
| 2 | 0 | 7.4e-09 | −0.283333 | −0.1 | −0.333334 | −0.083333 |
| **5.5** | **0** | **0.027614** | **−0.4** | **−0.033333** | **−0.45** | **3.9e-07** |
| **Outcome variables MPI effect of IPM** | | | | | | |
| 1 | 1.1e-12 | 1.1e-12 | −0.116667 | −0.116667 | −0.15 | −0.083333 |
| 2 | 0 | 0.000558 | −0.216666 | −0.05 | −0.283334 | −0.016667 |
| **2.7** | **0** | **0.037504** | **−0.266667** | **−0.016667** | **−0.316667** | **2.8e-07** |

## Recommendations

- Strengthen Resilience: These findings highlight the importance of promoting CSA practices as an effective pathway for reducing multidimensional poverty among rural households

- Promote Higher-Intensity CSA Adoption: Since poverty reduction outcomes are strongest among households who adopted CSA practices, programs should move beyond pilot activities to ensure broad, sustained adoption across communities, with support for training, inputs, and credit.

- Future studies should focus on accounting for endogeneity and feedback loops between technology adoption and poverty outcomes to capture the effect of poverty on CSA adoption.

**Table 10. Two stage least square (2SLS) model output.**

| MPI score (outcome variable) | Coefficient | Std. Err. | P>z |
|---|---|---|---|
| **HYV (treatment variable)** | −.1471345*** | .0450631 | 0.001 |
| Area | −.1484687*** | .0217406 | 0.000 |
| Agroecology | −.0141885 | .015187 | 0.350 |
| Sex | .0046432 | .0158653 | 0.770 |
| Marital status | −.0277207** | .0133337 | 0.038 |
| Education | .0011372 | .002306 | 0.622 |
| Age | −.0051056 | .0063504 | 0.421 |
| Age2 | .0000632 | .0000702 | 0.368 |
| TLU | .0020789 | .0025555 | 0.416 |
| Labor | −.0007673** | .0003165 | 0.015 |
| Total farm size | .0269427*** | .0072225 | 0.000 |
| Total income | −.0000195*** | 2.19e-06 | 0.000 |
| cons | 1.158031*** | .1423797 | 0.000 |
| **MPI score (outcome variable)** | **Coefficient** | **Std. Err.** | **P>z** |
| **SWC (treatment variable)** | −.2528677*** | .0814086 | 0.002 |
| Area | −.1545883*** | .0227003 | 0.000 |
| Agroecology | −.0074065 | .0178536 | 0.678 |
| Sex | .0088454 | .0183749 | 0.630 |
| Marital status | −.04387*** | .0159185 | 0.006 |
| Education | .0036243 | .0027335 | 0.185 |
| Age | −.0038078 | .0073098 | 0.602 |
| Age2 | .0000502 | .0000809 | 0.535 |
| TLU | .001712 | .0029509 | 0.562 |
| Labor | −.0009264** | .000364 | 0.011 |
| Total farm size | .0291794*** | .0084595 | 0.001 |
| Total income | −.0000191*** | 2.53e-06 | 0.000 |
| cons | 1.243923*** | .1676482 | 0.000 |
| **MPI score (outcome variable)** | **Coefficient** | **Std. Err.** | **P>z** |
| **spacing (treatment variable)** | −.2544787*** | .0779567 | 0.001 |
| Area | −.1536873*** | .0227646 | 0.000 |
| Agroecology | −.0180189 | .0176622 | 0.308 |
| Sex | .0123399 | .0189262 | 0.514 |
| Marital status | −.0160766 | .0162178 | 0.322 |
| Education | .0018242 | .0027062 | 0.500 |
| Age | −.0061706 | .0074632 | 0.408 |
| Age2 | .0000713 | .0000824 | 0.387 |
| TLU | −.0000468 | .0031594 | 0.988 |
| Labor | −.0009553*** | .0003726 | 0.010 |
| Total farm size | .0275629*** | .0084384 | 0.001 |
| Total income | −.000018*** | 2.66e-06 | 0.000 |
| cons | 1.256071*** | .1715335 | 0.000 |
| **MPI score (outcome variable)** | **Coefficient** | **Std. Err.** | **P>z** |
| **Crop rotation (treatment variable)** | −.152436*** | .040342 | 0.000 |
| Area | −.1562303*** | .0192205 | 0.000 |
| Agroecology | −.0149492 | .0152714 | 0.328 |

*(Continued)*

**Table 10.** (Continued)

| MPI score (outcome variable) | Coefficient | Std. Err. | P>z |
|---|---|---|---|
| Sex | .0076149 | .0160895 | 0.636 |
| Marital status | −.0224391* | .0136051 | 0.099 |
| Education | .0005855 | .0023397 | 0.802 |
| Age | −.0084854 | .0064772 | 0.190 |
| Age2 | .000102 | .0000715 | 0.154 |
| TLU | .0023017 | .0025594 | 0.368 |
| Labor | −.0007562** | .0003197 | 0.018 |
| Total farm size | .0270435*** | .0072283 | 0.000 |
| Total income | −.00002*** | 2.18e-06 | 0.000 |
| cons | 1.23577*** | .1464581 | 0.000 |
| **MPI score (outcome variable)** | **Coefficient** | **Std. Err.** | **P>z** |
| **Irrigation (treatment variable)** | −.996119* | .5316671 | 0.061 |
| Area | −.1785858*** | .0323248 | 0.000 |
| Agroecology | .0072097 | .0333347 | 0.829 |
| Sex | .0151193 | .0321114 | 0.638 |
| Marital status | .0307086 | .0417534 | 0.462 |
| Education | .0110768 | .0068314 | 0.105 |
| Age | −.0017896 | .0126264 | 0.887 |
| Age2 | −.0000175 | .0001456 | 0.904 |
| TLU | .0072344 | .0052814 | 0.171 |
| Labor | −.000801 | .0006219 | 0.198 |
| Total farm size | .0209403 | .0136677 | 0.125 |
| Total income | −.0000196*** | 4.28e-06 | 0.000 |
| cons | 1.029698*** | .2841021 | 0.000 |
| **MPI score (outcome variable)** | **Coefficient** | **Std. Err.** | **P>z** |
| **Crop diversification (treatment variable)** | −.1222782*** | .0354937 | 0.001 |
| Area | −.165749*** | .0178794 | 0.000 |
| Agroecology | −.0204649 | .0146978 | 0.164 |
| Sex | .008035 | .0156365 | 0.607 |
| Marital status | −.0249901* | .013117 | 0.057 |
| Education | −.0000341 | .0022982 | 0.988 |
| Age | −.0087402 | .0063037 | 0.166 |
| Age2 | .0001071 | .0000697 | 0.124 |
| TLU | .0038605 | .0024538 | 0.116 |
| Labor | −.0008197*** | .0003091 | 0.008 |
| Total farm size | .0235269*** | .0068537 | 0.001 |
| Total income | −.0000193*** | 2.15e-06 | 0.000 |
| cons | 1.207196*** | .1410483 | 0.000 |
| **MPI score (outcome variable)** | **Coefficient** | **Std. Err.** | **P>z** |
| **IPM (treatment variable)** | −.4569242*** | .1675807 | 0.006 |
| Area | −.121709*** | .0358308 | 0.001 |
| Agroecology | −.014498 | .0217335 | 0.505 |
| Sex | .0268604 | .0247217 | 0.277 |
| Marital status | −.0298824 | .0191303 | 0.118 |
| Education | .0064673* | .0037777 | 0.087 |
| Age | −.0149542 | .0098121 | 0.127 |

*(Continued)*

**Table 10.** (Continued)

| MPI score (outcome variable) | Coefficient | Std. Err. | P>z |
|---|---|---|---|
| Age2 | .0001688 | .0001075 | 0.116 |
| TLU | .004355 | .0036098 | 0.228 |
| Labor | −.0013937*** | .0004977 | 0.005 |
| Total farm size | .0353581*** | .0113786 | 0.002 |
| Total income | −.0000157*** | 3.61e-06 | 0.000 |
| cons | 1.474593*** | .2410756 | 0.000 |

Source: Model result, 2024.

**Table 11.** GSEM regression analyses.

| MPI (categorical) | Coefficient | Std. Err. |
|---|---|---|
| **Independent variable** | | |
| Total farm size | 0.0876123 | 0.064131 |
| Education | −0.0051601 | 0.0224154 |
| Total income | −0.0000572*** | 0.00002 |
| Area | −0.4690562*** | 0.1552767 |
| Sex | 0.0267309 | 0.1567544 |
| Marital status | −0.2676608** | 0.1233986 |
| Age | −0.1192106* | 0.0633536 |
| Age2 | 0.0013541* | 0.0007045 |
| Agroecology | −0.0809196 | 0.1451989 |
| Labor (ADeq) | −0.0071064** | 0.0031601 |
| Livestock (TLU) | 0.0050084 | 0.0247233 |
| **Treatment variable** | | |
| HYV | −0.728471*** | 0.1864136 |
| SWC | −0.6015962*** | 0.1914361 |
| Spacing | −0.6503183*** | 0.1792001 |
| Crop rotation | −0.6554536*** | 0.1859323 |
| Irrigation | −0.1672021 | 0.2500056 |
| Crop diversification | −0.7924444*** | 0.1866686 |
| IPM | −0.5338136*** | 0.1694423 |
| **MPI group** | | |
| /cut1 | −8.439512*** | 1.496998 |
| /cut2 | −8.329911*** | 1.494774 |
| /cut3 | −6.316684*** | 1.463269 |
| AIC | | 504.4241 |
| BIC | | 589.0179 |

Source: Model result, 2024.

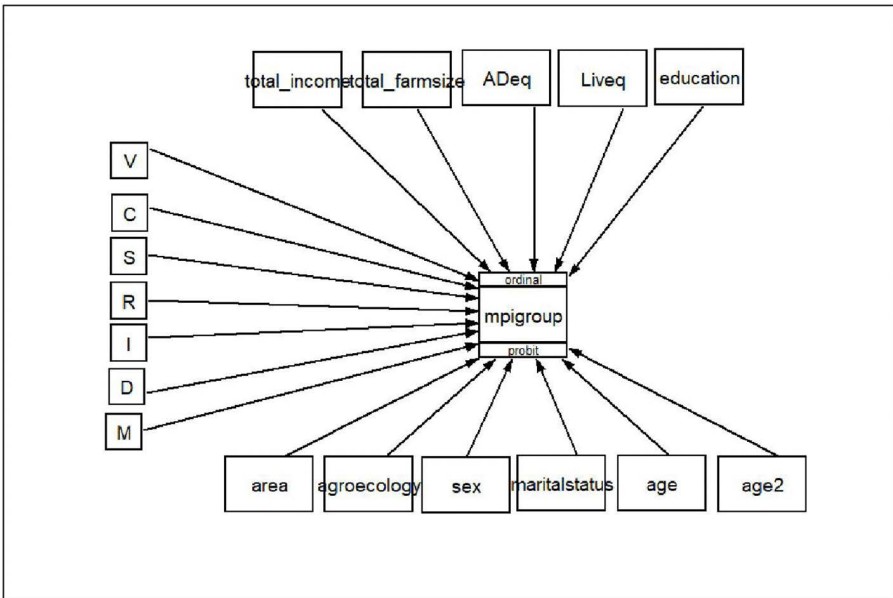

**Fig 4. Model GSEM.** Source: Model result, 2024.

## Author contributions

Conceptualization: Jemil Yasin Shifa, Abrham Seyoum Tsehay.

Formal analysis: Jemil Yasin Shifa.

Investigation: Jemil Yasin Shifa.

Methodology: Jemil Yasin Shifa.

Project administration: Jemil Yasin Shifa, Abrham Seyoum Tsehay.

Resources: Abrham Seyoum Tsehay.

Supervision: Abrham Seyoum Tsehay.

Validation: Abrham Seyoum Tsehay.

Visualization: Jemil Yasin Shifa, Abrham Seyoum Tsehay.

Writing – original draft: Jemil Yasin Shifa.

Writing – review & editing: Jemil Yasin Shifa, Abrham Seyoum Tsehay.

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
